# Positive Verbal Rewards, Creative Self-Efficacy, and Creative Behavior: A Perspective of Cognitive Appraisal Theory

**DOI:** 10.3390/bs13030229

**Published:** 2023-03-06

**Authors:** Zhenglin Liang, Sid Suntrayuth, Xiaohong Sun, Jiafu Su

**Affiliations:** 1Fine Arts College, Guangxi Arts University, Nanning 530022, China; 2International College, National Institute of Development Administration, Bangkok 10240, Thailand; 3School of Economics and Management, Wenzhou University of Technology, Wenzhou 325035, China; 4International College, Krirk University, Bangkok 10220, Thailand

**Keywords:** cognitive appraisal theory, positive verbal rewards, members’ creative behavior, positive affectivity, intermediate adjustment effect

## Abstract

The creative behavior of cultural innovation team members is the key to driving the team forward. Previous studies have relatively neglected the dynamic mechanism of positive verbal rewards on the creative behavior of cultural innovation team members. This paper, drawing on cognitive appraisal theory, focuses on the dynamic mechanism of positive verbal rewards on the creative behavior of cultural innovation team members and the moderating effect of positive affectivity. Based on the survey of 211 cultural innovation team members in Guangdong, China, this paper constructs a relationship model between positive verbal reward, creative self-efficacy, positive affectivity, and members’ creative behaviors and empirically tests the model. The results of statistical data analysis show that positive verbal reward has a significant positive impact on members’ creative behavior. Creative self-efficacy partially mediates between positive verbal rewards and members’ creative behavior; positive affectivity moderates the positive influence of creative self-efficacy on members’ creative behavior. The higher the level of positive affectivity, the stronger the positive impact of positive verbal rewards on members’ creative behavior, and vice versa. The above research findings help clarify the mechanism of positive verbal rewards on the cultural innovation team members’ creative behaviors in the context of Chinese organizations and provide theoretical support for cultural innovation team management practices.

## 1. Introduction

Creative industries with knowledge and culture as their core have gradually emerged around the world since the British government formally proposed the “creative economy” in 1998. Numerous data show that creative industries are becoming a new engine of economic development in many regions of the world. For example, Skavronska [1] stated that the rapid development of creative industries in Ukraine played a key role in the growth of the local economy. Yum [2] empirically investigated multiple locations such as California, Washington, and Los Angeles and found that creative industries were a major driver of urban economies in the United States. Jin and Li [3] pointed out that the output value of the UK creative industries increased by 53.1% between 2010 and 2017, nearly twice as fast as the country’s overall economic growth. In this context, the Japanese government has launched several policies to support creative industries since 2010, such as the New Growth Strategy, and has been improving a series of intellectual property laws and regulations, aiming to provide a favorable environment for the development of creative industries. China also included the cultural creative industry in the 14th five-year plan of national economic and social development in 2021 to enhance the quality of development of the cultural creative industry through policy support. It is evident that creativity is an important driving force of economic development and the key to the high-quality development of the cultural creative industry. Creativity has the characteristics of individual voluntary spontaneity and inseparability from the provider of creativity, and cannot exist independently from the individual who provides the creativity, as it must be embodied by the innovative ideas and creative behavior of the members of the cultural innovation team in the company. Therefore, exploring the influence of a reward system on members’ creative behaviors in cultural innovation teams is important for the high-quality development of the cultural creative industry [4].

Cultural innovation teams need continuous innovation to meet the demands of constant product upgrades. The attitudes and behaviors of team members determine the quality of creativity and the effectiveness of the product. Thus, the creative behavior of team members is the basis and inexhaustible motivation for the sustainable development of cultural innovation teams, and ultimately determines their competitive advantage. An effective reward system to stimulate creative behavior of members is related to the success or failure of cultural innovation teams. Some scholars have focused on the influencing role of external rewards on intrinsic motivation [5]. However, what external rewards are effective in motivating intrinsic motivation and enabling behaviors to be performed has generated a wide range of current scholarly debates. Hennessey and Amabile [6] argued that material rewards facilitated team members’ intrinsic motivation and creative behavior. Deci et al. [7] showed that material rewards can impair creativity, cognition, and problem-solving skills associated with intrinsic motivation. Malek et al. [8] also confirmed that material rewards made members feel that creative behavior was interfering, and may therefore weaken intrinsic motivation. Cameron and Pierce (1994) [9] found that among external rewards, positive verbal rewards (PVRs) promoted intrinsic motivation and work engagement. In terms of cognitive appraisal theory, first, self-determination and self-efficacy are two key psychological needs that influence intrinsic motivation and behavior, and members’ intrinsic motivation is enhanced if the form of reward can effectively meet the psychological needs [10]. Second, there are two general types of external reward forms that influence intrinsic motivation: controlled and informational [11]. These material rewards were typically used to persuade people to do things they would not originally do, which tended to make members feel that creative behavior was interfered with and controlled by the organization, which could reduce autonomy and undermine intrinsic motivation. In contrast, positive verbal rewards are considered to be informative and positive feedback because they tend to be closely related to the behavioral activity itself and allow members to feel support from the organization, thus better explaining the contribution of external rewards to members’ intrinsic motivation and behavior [12]. Since the atmosphere created by positive verbal rewards makes members feel valued by the organization for their creativity, their creative behavior also increases significantly. Therefore, this study concludes that positive verbal rewards in cultural innovation teams can drive more members’ creative behaviors.

Social cognitive theory notes that self-beliefs exert a vital impact on individual behavior. In cultural innovation teams, members’ risk assessment of innovative creative activities and confidence in their innovation success are closely related to their creative behaviors [13,14]. Positive verbal rewards are an important way to enhance creative self-efficacy, and the efficiency of the execution of creative behaviors is enhanced when members have high self-confidence in their creative success. Xing et al. [15] indicated that although positive verbal rewards contributed to the development of creative self-efficacy, it did not directly trigger creative behavior. Therefore, this study hypothesizes that positive verbal rewards have an impact on members’ creative behaviors by influencing their creative self-efficacy.

In team practice, even members who also receive verbal encouragement can vary in their perceived level of moral support and the level of self-confidence subsequently triggered [16]. In the view of Raftery and Bizer [17], the emergence of this phenomenon is closely related to the personality–emotional characteristics of the members. Positive affectivity is the tendency of individuals to respond positively to environmental stimulus, and is a personality emotion related to self-efficacy [18]. Selmer and Lauring’s [19] study showed that members with higher levels of positive affectivity tend to perceive team feedback positively, and think and act in ways that support positive affectivity. This state of optimism helps to reinforce the relationship between creative self-efficacy and members’ creative behaviors [20]. Therefore, this study suggests that positive affectivity will play a moderating role in the relationship between creative self-efficacy and members’ creative behaviors. 

This paper contributes to the study of the motivation of creative behavior of cultural innovation team members from several perspectives. First, this study expands our understanding of possible antecedent variables of creative behavior and possible outcomes of positive verbal rewards by exploring the effects of positive verbal rewards on members’ creative behaviors. Second, this study analyzed the possible mediating (creative self-efficacy) and moderating (positive affectivity) effects between positive verbal rewards and members’ creative behaviors, which helped us to understand in greater detail the effects and mechanisms of positive verbal rewards on members’ creative behaviors. Moreover, we proposed a path model for explaining the relationship between positive verbal rewards and members’ creative behaviors based on cognitive appraisal theory (see Figure 1). Third, while previous studies have focused on the effects of external rewards such as material and monetary rewards based on members’ creative behaviors, this study proposes that managers should focus on promoting team members’ creative behaviors through moral support and positive feedback from the perspective of positive verbal rewards, which will provide more valuable references for the practice of managing creative behaviors in cultural innovation teams.

Throughout the existing studies, it can be found that current researchers are increasingly focusing on the motivation and promotion of team member’s creative behaviors in various industries [21,22,23]. However, most studies on individual creative behaviors in the Chinese context are conducted on students, team leaders, or non-cultural creative industry practitioners, and few studies focus on team members or even cultural innovation team members as a group [24,25,26]. Motivating team members at the mental level is one of the important means for team managers to promote their members’ creative behaviors, but the effectiveness of this needs to be confirmed by further research. Therefore, this study constructs a theoretical model based on cognitive appraisal theory, with positive verbal rewards as the independent variable, creative self-efficacy as the mediating variable, members’ creative behaviors as the dependent variable, and positive affect as the moderating variable, to explore the formation mechanism of members’ creative behaviors in the context of Chinese cultural innovation teams, in order to provide a reference for the sustainable development of Chinese cultural innovation teams. The purpose of this study is to explore the formation mechanism of members’ creative behaviors in the context of Chinese cultural innovation teams, with a view to providing reference for the sustainable development of Chinese cultural innovation teams.

## 2. Related Work and Hypothesis Development

### 2.1. Positive Verbal Rewards and Members’ Creative Behavior

In cultural innovation teams, positive verbal rewards are a form of reward in which the team uses verbal or written verbal expressions to recognize members for good behavior [27]. Cognitive evaluation theory suggests that positive verbal rewards can be understood as supportive informational feedback that motivates individuals to engage in more creative activities [28]. Sidharta [29] defines team members’ creative behavior as the behavior of individuals who put new ideas into practice by coming up with new concepts, developing new design techniques and methods, and new cultural products and activities based on the purpose of improving the quality of the team’s output and the level of creativity. The creative behavior can be manifested in three major ways: the generation, promotion, and implementation of creative thinking [30]. Robescu et al. [31,32] have shown that cultural innovation teams show support and attention to members’ creative behaviors through verbal forms of encouragement, approval, and praise, and it can make members feel affirmed and accepted from the organization, which will contribute to the stabilization, continuity, and longevity of their creative behaviors. Therefore, this study proposes the following hypotheses:

**H1:** 
*Positive verbal rewards positively promote members’ creative behaviors.*


### 2.2. The Mediating Role of Creative Self-Efficacy

Creative self-efficacy refers to the degree of self-efficacy demonstrated by members engaging in creative activities [33]. It is mainly expressed in four aspects: members believing in their ability to generate novel ideas, believing in their ability to solve problems creatively, having the willingness to help others to accomplish creative tasks, and having confidence in their ability to apply new methods to solve existing problems [34]. Before engaging in creative activities, individuals usually determine whether they have the ability to conceive of ideas and the extent to which they can bring them to life before they perform the corresponding behavior [35]. One of the important reasons why some members have good creative potential but do not further manifest it in their actions is that they lack sufficient knowledge and self-confidence to support their ability to successfully realize their ideas. Positive verbal feedback is an effective way to strengthen creative self-efficacy, especially when team members are faced with complex and challenging tasks, appropriate verbal encouragement and supportive attitude, which can effectively enhance members’ confidence to overcome difficulties. The available literature argued that since members may lack sufficient confidence to measure their success in achieving creativity, team managers expressing trust in their members through words, affirming their new ideas, and showing support for their implementation plans will help shape members’ creative self-efficacy and enhance their self-confidence in achieving creativity [36,37,38]. Therefore, this study proposes the following hypotheses:

**H2:** 
*Positive verbal rewards positively contribute to creative self-efficacy.*


Creative self-efficacy is an important influencing factor for members to decide whether to engage in creative behavior. Compared with the more repetitive routine work, creative work often implies more challenges and difficulties, more time and effort, and thus requires positive and strong self-beliefs to support individuals to overcome the difficulties and eventually complete the creative tasks. Newman et al. [39,40] noted that high levels of creative self-efficacy contribute to creative behavior. Conversely, members at low levels of creative self-efficacy misjudge the riskiness of problems due to their perceived lack of innovative thinking and creative problem-solving skills [41]. In cultural innovation teams, members with higher creative self-efficacy tend to be more likely to be curious about new things, more willing to actively explore rather than passively accept, to try new technologies and methods, and to be more open and proactive about new knowledge and more willing to use non-traditional approaches to problem solving and thus engage in creative behavior. In this regard, this study proposes the following hypotheses: 

**H3:** 
*Creative self-efficacy contributes positively to members’ creative behavior.*


After reviewing the aforementioned literature, it can be seen that many scholars have discussed the direct impact of positive verbal reward on members’ creative behavior from the perspective of positive verbal reward, but the research on the mediating effect of creative self-efficacy on the relationship between the two is relatively rare. Creative self-efficacy is essentially a member’s awareness and identification of their own creative ability. Team managers provide members with messages of appreciation and support through positive verbal rewards so that members can feel a sense of accomplishment, trust, and security in their work, and thus creative self-efficacy and creative behavior are reinforced accordingly. Jaussi et al. have shown that creative self-efficacy plays an important mediating role in the creative activity of cultural innovation [42]. Tierney and Farmer [43] stated that the team’s expectation of creativity from its members increases the probability that they will adopt creative behaviors. However, the team’s creativity expectations do not directly affect the members’ creative behaviors, but rather they firstly show approval and support to the members’ creative behaviors through positive verbal encouragement, and then indirectly influence the employees’ creative behaviors through the members’ creative self-efficacy. Therefore, this work concludes that positive verbal rewards can enhance members’ creative self-efficacy, and also creative self-efficacy has a positive effect on members’ creative behaviors [44,45,46], and makes the following assumptions:

**H4:** 
*Creative self-efficacy mediates the relationship between positive verbal rewards and members’ creative behaviors.*


### 2.3. Moderating Effect of Positive Affectivity

Watson et al. [47] defined positive affectivity as the degree to which an individual feels enthusiastic, active, and alert, and stated that positive affectivity implies that the individual is in a pleasant, active, highly focused, and engaged state. Studies have confirmed that optimistic and positive employees are more engaged and creative [48]. When working in a good emotional state, members are more likely to come up with new ideas, try new approaches, and be more flexible in solving problems or achieving goals. Based on the positive affectivity extension-construction theory, Fredrickson [49] argues that positive affectivity expands individuals’ cognitive range and motivates them to pursue novel and creative paths of thought and action, thus, the level of members’ positive affectivity can lead to differences in the effectiveness of creativity. Wang and Deng [50] also found that positive affectivity played a moderating influence in promoting members’ creative behaviors, and suggested that positive affectivity enhanced or slowed down individuals’ behaviors by influencing members’ creative self-efficacy. This paper concludes that the effect of creative self-efficacy on members’ creative behaviors may be moderated by the role of positive affectivity. Even members who have the same level of self-confidence in accomplishing creative work may differ in the performance or effectiveness of creative behaviors depending on the level of positive affectivity state they are in. A member in a high level of positive affectivity state experiences pleasure and pride when engaging in creative activities, which makes them more likely to be inspired to innovate, more confident to explore the feasibility of ideas, and more likely to produce more creative behaviors [51]. Thus, high levels of positive affectivity contribute to the perception of creative self-efficacy and will lead to more creative actions from creative self-efficacy. On the contrary, members in a low level of positive affectivity state, even though they are confident enough in their success in achieving creative actions, are not sufficiently engaged and easily distracted in conceiving ideas, which will make it difficult for them to catch a flash of inspiration and eventually lead to a low level of innovation. Therefore, this paper concludes that positive affectivity positively moderates the effect of creative self-efficacy on members’ creative behaviors and that the interaction between positive affectivity and creative self-efficacy has a significant positive effect on members’ creative behaviors, and gives the following assumptions:

**H5:** 
*Positive affectivity moderates the relationship between creative self-efficacy and members’ creative behaviors.*


Based on the above statements, this study constructed the following model (see Figure 1).

## 3. Research Design

### 3.1. Procedures and Participants

The cultural innovation team studied in this paper specifically refers to the professional team engaged in creative activities in the environment of cultural and creative industries. Accordingly, this study mainly selects individuals who work creatively in a collaborative team approach with the cultural and creative industries as the research objects. At the same time, considering the essential attributes of the cultural innovation team, the sample objects should also meet the following conditions:(a)The team is mainly composed of professionals engaged in creative planning, product development or related technical services;(b)The team has the technical equipment and workplace required for creative planning, product development, and corresponding business;(c)The team has self-developed, produced, or owned its own intellectual property products as well as explanatory materials;(d)The team has a good working atmosphere.

The respondents to the formal questionnaire were mainly from the creative design departments and product design departments of cultural and art institutions, design service companies, and cultural and creative enterprises in Guangdong province, China, who are engaged in the research and development of cultural and creative products and the planning of related activities. The questionnaires were collected online and distributed in cities of Guangzhou, Shenzhen, Foshan, Zhuhai and Zhongshan in Guangdong province, China, mainly through two channels: field research by enterprises and forwarding by commissioned acquaintances. A total of 261 questionnaires were distributed in this study and 248 were returned, with a return rate of 95%. Among them, 25 questionnaires had the problem that all the options were the same or the options were logically contradictory, and 12 questionnaires had too short a response time, so a total of 37 questionnaires were excluded due to the above problems. The final number of valid questionnaires obtained in this study was 211, and the effective rate was 85.1%. Among all the subjects, 52.1% were male and 47.9% were female, 70.2% were under 35 years old, and 88.6% had a bachelor’s degree or above. The subjects involved in a variety of cultural and creative industries, animation accounted for 30.3%, advertising accounted for 24.6%, environmental art accounted for 16.1%, product design accounted for 12.3%, film and television media accounted for 11.4%, and others accounted for 5.2%. Table 1 summarizes the demographic information of the samples in this study.

### 3.2. Variable Measurement

This study used a questionnaire to test the hypothesis. The questionnaire was prepared on the basis of compiling the literature on positive verbal rewards, creative self-efficacy, creative behaviors and positive affectivity, and making full reference to existing scales. In addition, three scholars in the field of cultural and creative industry research and three industry veterans were invited to revise the wording and improve the language of the questionnaire content. The formal questionnaire consists of two parts: respondents’ personal information and a study on the influence of positive verbal rewards on members’ creative behaviors. The second part of the scale was measured by a Likert scale.

Four variables were involved in this study: positive verbal rewards, creative self-efficacy, members’ creative behaviors, and positive affectivity. All scales were measured by a Likert five-point scale, and scored 1, 2, 3, 4, and 5 on a scale of “very non-conforming”, “relatively non-conforming”, “average”, “relatively conforming”, and “very conforming”, respectively, and the scores of each construct were totaled by the mean value.

Positive verbal reward. This study was adapted from a scale developed by Andersen et al. [52] with three questions. They are, “I get positive feedback from the team when I come up with new ideas”, “The team shows appreciation for me when I complete a task in a new way and get better results than expected”, and “The team shows appreciation for me when I am recognized or encouraged by the team when I perform creatively”. The Cronbach’s alpha coefficient for this scale was 0.836.

Creative self-efficacy. The scale used in this study was adapted from the scale developed by Carmeli and Schaubroeck [34] and consists of eight questions. The scale items included “I can accomplish task goals in a refreshing way” and “I am confident that I can achieve task goals differently and with better results”, among others. The Cronbach’s alpha coefficient for this scale was 0.908.

Members’ creative behaviors. Based on the work characteristics of cultural innovation teams, this study appropriately semantically revised the findings of Scott and Bruce [53] by asking respondents to assess their own performance of creative behaviors at work with six question items. For example, “I often use creative approaches to problem solving”. The Cronbach’s alpha coefficient for this scale was 0.879.

Positive affectivity. This study followed the positive affectivity Scale developed by Tang et al. [54], which was extracted and adapted from Watson et al.’s [47] PANAS Affect Scale to be more relevant to the Chinese cultural context and the comprehension habits of cultural innovation team members, with five question items including, for example, “I am passionate about my work”. The Cronbach’s alpha coefficient for this scale was 0.857.

Control variables. In this study, gender, age, education, and industry were set as control variables. ① Gender: it has been suggested that males and females may differ in reward perception and motivation [55]. Therefore, in this study, gender was included in the measurement, and “1” was defined as male and “2” as female. ② Age: the age of team members reflects to a certain extent the individual’s creative potential, as well as their ability to learn and accept new things. Therefore, this study uses statistical methods to observe the age distribution structure of members, defining “1” as 25 years old and below, “2” as 26–35 years old, “3” as 36–45 years old, and “4” as 46 years old and above. 45 years old, and “4” represents 46 years old and above. ③ Education, i.e., the degree of education received by individuals, is one of the important observation indicators affecting individual creative self-efficacy and creative activity. In this study, “1” represents college degree or below, “2” represents bachelor’s degree, “3” represents master’s degree, and “4 “ represents doctoral degree and above. ④ Industry: innovation teams from different cultural industries differ in their values, which may have an impact on members’ perceptions of rewards, creative self-efficacy, and creative behaviors. Therefore, this study includes industry in the scope of observation, using “1” for advertising media, “2” for film and media, “3” for product design, “4” for animation design, “5” for environmental art design, and “6” for others.

## 4. Results Analysis

### 4.1. Validation Factor Analysis

To examine the discriminant validity of all variables, this study conducted a validated factor analysis based on Zhou and Long’s [56] common method biases statistical approach using SPSS 26.0 for the four variables involved in this study: positive verbal rewards, creative self-efficacy, members’ creative behaviors, and positive affectivity. The results are shown in Table 2, and the four-factor model had the best fit with χ^2^ = 328.565, df = 203, χ^2^/df = 1.619, RMSEA = 0.054, SRMR = 0.041, CFI = 0.948, and TLI = 0.941, providing support for the discrimination involving the four variables in this study.

### 4.2. Descriptive Statistics and Correlation Analysis

Before descriptive statistics and correlation analysis, the Harman single-factor method was first applied to test for common method bias: the first principal component variance contribution rate was 22.276%, which was below the critical value of 40%, indicating that there was no common bias problem. Based on the means, standard deviations, and correlation coefficients of the variables presented in Table 3, the correlation between the variables was significant. Positive verbal rewards positively regarding creative self-efficacy (r = 0.426, *p* < 0.01) and members’ creative behavior (r = 0.483, *p* < 0.01) and creative self-efficacy positively regarding members’ creative behavior (r = 0.532, *p* < 0.01) supported the hypotheses of this paper. Preliminary justification for the model is constructed in this paper.

### 4.3. Hypothesis Testing Analysis

(1)Main effect and mediating effect

The main effect and mediating effect were first considered, and H1, H2, H3, and H4 were tested by using SPSS 26.0 applying cascade regression: ① The control variables (gender, age, education, and industry) and independent variables (positive verbal rewards) were sequentially introduced into the regression equation to analyze the effect of positive verbal rewards on members’ creative behaviors. ② The control variables and independent variables (positive verbal rewards) were introduced, in turn, to analyze the effect of positive verbal rewards on creative self-efficacy. ③ Introduce control variables and mediating variable (creative self-efficacy) in turn to analyze the effect of creative self-efficacy on members’ creative behaviors. ④ Mediating effect and control variables were introduced first, and then independent variable and mediating variable were put in to analyze the effects of positive verbal rewards and creative self-efficacy on members’ creative behaviors. The results are shown in Table 4. According to Model 4, the positive effect of positive verbal rewards on members’ creative behavior was significant (β = 0.472, *p* < 0.01). Thus, H1 was verified. In Model 2, the positive effect of positive verbal rewards on creative self-efficacy was significant (β = 0.410, *p* < 0.01). Thus, H2 was validated. Model 5 indicated a significant positive effect of creative self-efficacy on members’ creative behaviors (β = 0.534, *p* < 0.01), and H3 was validated. In Model 6, after adding creative self-efficacy, the positive effect of positive verbal rewards on members’ creative behaviors was still significant (β = 0.305, *p* < 0.01), but the coefficient was reduced, while the positive effect of creative self-efficacy on members’ creative behaviors was significant (β = 0.407, *p* < 0.01). Thus, H4 was verified that creative self-efficacy partially mediates the relationship between positive verbal rewards and members’ creative behaviors. The test for mediating effects was performed using the Bootstrap method, and the non-parametric percentage Bootstrap was corrected by deviation while using the bias-corrected test with 2000 replicate samples at 95% confidence intervals. According to the criterion for judging the mediation effect proposed by Wen and Ye (2014), a significant mediation effect is indicated if the variable takes a value interval that does not include 0 at the 95% confidence interval. As can be seen from Table 5, the standardized indirect effect value of positive verbal rewards → creative self-efficacy → members’ creative behaviors is 0.158, the confidence interval of the indirect effect does not contain 0, and the *p*-value is less than 0.01, which shows that creative self-efficacy plays a mediating role between positive verbal rewards and members’ creative behaviors (See Figure 2).

(2)Moderating effect

In order to test whether there is an interaction between creative self-efficacy and positive affectivity, the control variables (gender, age, education, and industry affiliation), the mediating variable (creative self-efficacy) and the moderating variable (positive affectivity) were introduced into the regression equation on the basis of setting members’ creative behaviors as the dependent variables, and finally the interaction term between creative self-efficacy and positive affectivity was put in. The results of the hierarchical regression analysis derived from applying the Process plug-in of SPSS 26.0 are shown in Table 4.

According to Model 8 in Table 3, the interaction term between creative self-efficacy and positive affectivity has a significant positive effect on members’ creative behaviors (β = 0.245, *p* < 0.01), which indicates that positive affectivity moderates the relationship between creative self-efficacy and members’ creative behaviors: when the level of positive affectivity is higher, the greater the positive effect of creative self-efficacy on members’ creative behaviors, while when the level of positive affectivity is low, the positive influence of creative self-efficacy on members’ creative behaviors decreases, so hypothesis H5 is validated.

Figure 3 shows the moderating effect of positive affectivity on creative self-efficacy and members’ creative behaviors. This study depicts the interaction between creative self-efficacy and positive affectivity based on the moderating effect analysis method of Fang et al. [43] with one standard deviation above the mean and one standard deviation below the mean, respectively. The moderating effect value was 0.659 (*p* < 0.01) for high levels of positive affectivity and 0.306 (*p* < 0.01) for low levels of positive affectivity, which shows that the degree of influence of creative self-efficacy on members’ creative behaviors is stronger at high levels of positive affectivity states.

## 5. Discussion and Recommendation

### 5.1. Theoretical Contributions

Based on the results of the above empirical tests, this study concludes the following theoretical contributions:

First, in cultural innovation teams, positive verbal rewards have a significant positive effect on members’ creative behaviors, and when members receive more positive verbal rewards in the work process, their creative behaviors increase accordingly. The findings of this study support Cameron and Pierce’s [9] view to a certain extent in that they test the effectiveness of positive verbal rewards in motivating members to perform creative behaviors, enrich the research on positive verbal rewards in the form of external rewards, and improve the theoretical framework of positive verbal rewards influencing members’ creative behaviors.

Second, in cultural innovation teams, creative self-efficacy mediates members’ creative behaviors under the influence of positive verbal rewards. Most previous studies have explored the role of leadership style on creative self-efficacy [57], or analyzed the factors influencing creative behavior in terms of identity, environmental influence, and willingness to take on [58,59,60], and less often have intrinsic motivational elements including variables such as “creative self-efficacy” embedded. This study constructs the evolutionary path of “positive verbal reward—creative self-efficacy—members’ creative behavior” and draws conclusions, which not only enrich the research on the antecedent variables of creative self-efficacy, but also expand the research on the outcome variables of creative self-efficacy, and provide theoretical support for further research on the psychogenesis of the relationship between positive verbal reward and members’ creative behavior.

Third, positive affectivity can positively regulate the relationship between creative self-efficacy and members’ creative behaviors. The interaction between positive affectivity and creative self-efficacy shows that members with higher levels of positive affectivity also have higher perceptions of creative self-efficacy, which makes the effect of creative self-efficacy on creative behaviors stronger to some extent, and members’ creative behaviors more likely to be achieved at this time. Previous studies have mainly considered the moderating effect between creative self-efficacy and members’ creative behaviors from external situational variables, but less from the perspective of affective traits [61,62,63]. The results also coincided with Mielniczuk and Laguna [64] which found that self-efficacy predicts the innovative behavior of entrepreneurs. The findings of this study not only test the relationship between positive verbal rewards and members’ creative behaviors as a series of influential processes in which positive feedback satisfies individual psychological needs, which in turn stimulates intrinsic motivation and leads to behaviors, but also deepen the research on the influence of positive affectivity on the regulation of creative self-efficacy and provide ideas for further research on positive affectivity in the future.

Finally, there were no gender differences in the effects of positive verbal rewards on creative self-efficacy, which is inconsistent with Cohen’s [65] suggestion that positive verbal rewards positively affect only males.

### 5.2. Practical Implications

This study draws conclusions based on a study of positive verbal rewards and members’ creative behaviors in cultural innovation teams. This study broadens the scope of the research scenario of positive verbal rewards, deepens the understanding of cultural innovation team managers about the usefulness of positive verbal rewards, and facilitates cultural innovation teams to enhance members’ creative behaviors through positive verbal rewards. This study has good implications for cultural innovation teams trying to enhance team innovation development through spiritual level rewards.

First, the findings suggest that positive verbal rewards can achieve more creative behaviors by stimulating creative self-efficacy, and thus accomplish the team’s desired innovation goals. Therefore, in the case of Chinese cultural innovation team management, it is necessary for team managers to activate and maintain the enthusiasm of team members through diverse positive verbal incentives and to pay attention to verbal expressions when communicating: (1) managers use more positive motivational language when communicating with new inductees, which can effectively relieve new members’ tension and sensitivity, make them feel that the team recognizes their value, and enhance their confidence in carrying out their new work. (2) Managers should always pay attention to the dynamics and development of members, give positive incentives to members who put forward feasible creative proposals, help them build confidence in continuous innovation, and also solve the problems they encounter in their creativity in a timely manner, and use supportive words to cultivate their courage and confidence to make further efforts. (3) Use positive language to create a good collaborative communication atmosphere so that members can psychologically feel the acceptance and trust of the team, thus enhancing their sense of competence and participation in creative work.

Second, strengthening members’ creative self-efficacy level should also be a high priority for cultural innovation team managers in practice. On the one hand, managers can improve the creative self-efficacy of their members through various training means, such as systematic course training related to creative ideas, lectures on classic creative case studies, and sharing of successful experiences of team members to improve members’ sense of creative self-efficacy. On the other hand, managers should fully understand the degree of each member’s ability and provide members with moderately challenging work, while focusing on encouraging and supporting members through verbal expressions, so as to raise members’ work expectations and believe that they can produce better creative results than the expected goals.

Finally, managers should pay attention to the role of psychological state in influencing members’ creative behaviors. A friendly team working atmosphere and harmonious collegial cooperation often help members keep a positive and optimistic psychological state. Therefore, on the one hand, team managers should strengthen members’ self-mental management ability through psychological training and daily effective communication, and try their best to shape a positive and optimistic mentality. On the other hand, team-building activities should be carried out appropriately to help members maintain a positive and optimistic psychological state by cultivating a friendly team working atmosphere and a harmonious colleague cooperation relationship, so as to promote their creative behavioral performance in the workplace.

### 5.3. Limitations and Future Directions

This study explored the mediating role of creative self-efficacy between positive verbal rewards and members’ creative behaviors through a theoretical deductive and empirical analysis research method, and concluded that there is a moderating effect of positive affectivity on the relationship between creative self-efficacy and members’ creative behaviors. Despite the progress made in this study, there are still some limitations that need to be further explored in subsequent studies.

First, this study only investigated cultural innovation teams in Guangdong, China, and has not yet addressed members of cultural innovation teams in other regions, which to some extent limits the breadth of the study’s findings. A cross-regional comparative study could be considered in the future to explore the differences and connections between the effects of positive verbal rewards and members’ creative behaviors in different human geographic settings.

Second, this study only examined the moderating effects of positive affectivity and did not examine positive affectivity subdivision. In the future, we may consider exploring the moderating effects of positive affectivity with different motivational strengths between creative self-efficacy and creative behavior. 

## Figures and Tables

**Figure 1 behavsci-13-00229-f001:**
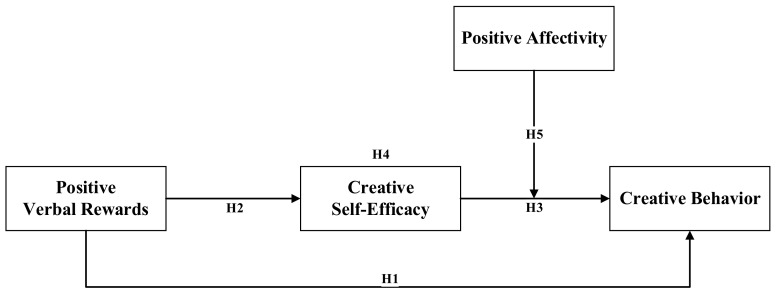
Research model for analyzing the occurrence of creative behaviors among cultural innovation team members from the perspective of positive verbal rewards.

**Figure 2 behavsci-13-00229-f002:**
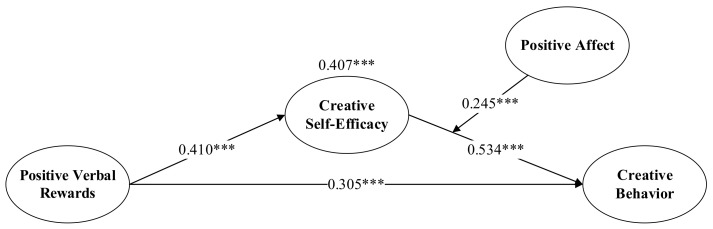
Model validation result, *** indicates *p* < 0.001.

**Figure 3 behavsci-13-00229-f003:**
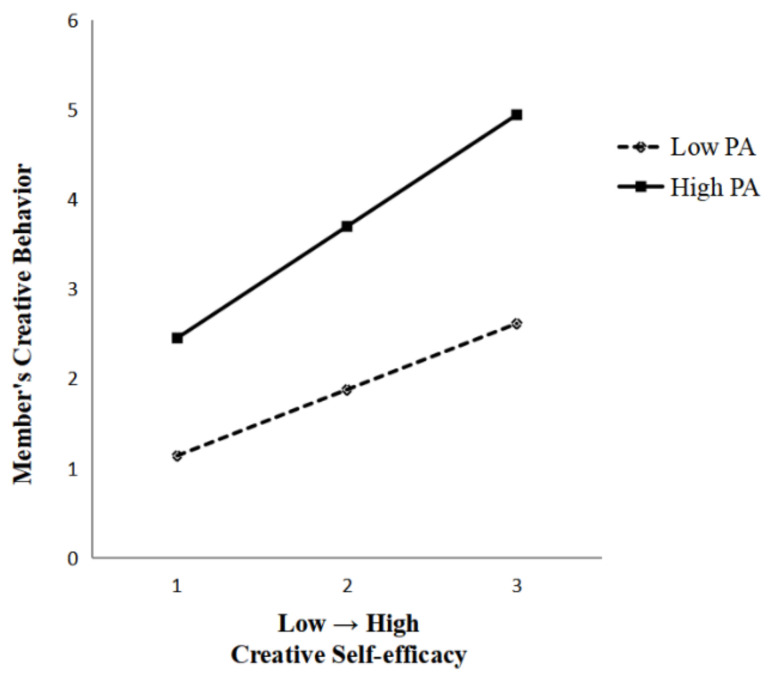
Interaction between creative self-efficacy and members’ creative behaviors.

**Table 1 behavsci-13-00229-t001:** Demographic characteristics of participants.

Variables	Category	Frequency	%
Gender	Male	110	52.1
	Female	101	47.9
Age	25 and below	43	20.4
	26–35	105	49.8
	36–45	54	25.6
	46 and above	9	4.3
Academic degree	Junior college and below	24	11.4
	Bachelor’s degree	155	73.4
	Master’s degree	28	13.3
	PhD or above	4	1.9
Industry	Advertising media	52	24.6
	Film and television media	24	11.4
	Product design	26	12.3
	Animation design	64	30.3
	Environmental art design	34	16.1
	Other	11	5.2

**Table 2 behavsci-13-00229-t002:** Results of validation factor analysis.

Model	Factor	χ^2^	*df*	χ^2^/*df*	RMSEA	SRMR	CFI	TLI
Model 1	PVR + CSE + MCB + PA	1191.196	209	5.700	0.150	0.108	0.592	0.549
Model 2	PVR + CSE, MCB, PA	833.714	208	4.008	0.120	0.087	0.740	0.712
Model 3	PVR + CSE + MCB, PA	527.816	206	2.562	0.086	0.064	0.866	0.850
Model 4	PVR, CSE, MCB, PA	328.565	203	1.619	0.054	0.041	0.948	0.941

Note: PVR = positive verbal reward, CSE = creative self-efficacy, MCB = member’s creative behavior, PA = positive affectivity.

**Table 3 behavsci-13-00229-t003:** Correlation analysis.

Variables	Average Value	Standard Deviation	1	2	3	4	5	6	7	8
1. Gender	1.480	0.501	1							
2. Age	2.140	0.784	−0.023	1						
3. Education	2.060	0.566	0.021	0.014	1					
4. Industry	3.180	1.580	−0.01	−0.043	0.037	1				
5. Positive verbal rewards	3.701	0.763	−0.015	−0.149 *	−0.056	−0.043	1			
6. Creative self-efficacy	3.924	0.706	0.053	−0.161 *	0.02	−0.117	0.426 **	1		
7. Members’ creative behaviors	4.009	0.721	0.013	−0.166 *	−0.03	0.049	0.483 **	0.532 **	1	
8. Positive affectivity	4.065	0.696	0.156 *	−0.141 *	−0.026	0.143 *	0.179 **	0.168 *	0.463 **	1

Note: * indicates *p* < 0.05, ** indicates *p* < 0.01.

**Table 4 behavsci-13-00229-t004:** Hypothesis test results.

	Category	Creative Self-Efficacy		Members’ Creative Behaviors					
		Model 1	Model 2	Model 3	Model 4	Model 5	Model 6	Model 7	Model 8
Control variables	Gender	0.047	0.054	0.010	0.018	−0.015	−0.004	−0.071	−0.053
	Age	−0.165 *	−0.104	−0.164 *	−0.093	−0.076	−0.051	−0.036	−0.014
	Education	0.026	0.047	−0.030	−0.005	−0.044	−0.025	−0.029	−0.016
	Industry	−0.124	−0.104	0.043	0.066	0.109	0.108	0.048	0.059
Independent variable	Positive verbal rewards		0.410 **		0.472 **		0.305 **		
Mediating variable	Creative self-efficacy					0.534 **	0.407 **	0.472 **	0.481 **
Moderating variable	Positive affectivity							0.381 **	0.384 **
Interaction effect	Creative self-efficacy × positive affectivity								0.245 **
*R^2^*		0.044	0.207	0.030	0.247	0.303	0.379	0.436	0.495
△*R^2^*		0.044	0.163	0.030	0.217	0.273	0.348	0.406	0.477
*F*		2.370	10.721 **	1.616	13.454 **	17.862 **	20.709 **	26.290 **	28.414 **

Note: * at the 0.05 level (two-tailed), the correlation is significant. ** at the 0.01 level (two-tailed), the correlation is significant.

**Table 5 behavsci-13-00229-t005:** Mediation effect test.

Path	Effect	Effect	BootSE	BootLLCT	BootULCT
Positive verbal rewards → Creative self-efficacy → Members’ creative behaviors	Total effect	0.446	0.093	0.255	0.632
	Direct effect	0.288	0.093	0.200	0.561
	Indirect effects	0.158	0.070	0.054	0.329

## Data Availability

The data used to support the findings of this study are included within the article.

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
