# Peer review of "Positive Verbal Rewards, Creative Self-Efficacy, and Creative Behavior: A Perspective of Cognitive Appraisal Theory"

_behavsci, 2023, doi:10.3390/bs13030229_

Round 1

Reviewer 1 Report

This paper is very well written! It gives the needed detail but yet it is still to the point. I truly don’t have a lot of constructive comments to give.

The main one I’m seeing it that many of your citations are older, which is fine because some are staple pieces. However, you want to demonstrate this is still being discussed and research on, meaning to you adding to a current conversation.

Here are two examples that will show there are still recent works to add to regarding creative self-efficacy:

This one discusses the usefulness of creative self-efficacy leading to team members viewing each other as a resource and greater team satisfaction:

Swab, R.G., Cogan, A., Pret. T., & Marshall, D.R. (2021). Examining the creative self-efficacy, goal interdependence, and satisfaction of new venture teams in the board game industry. Entrepreneurship Research Journal. doi.org/10.1515/erj-2021-0142

This one further supports your argument of creative self-efficacy and innovation:

Park, N. K., Jang, W., Thomas, E. L., & Smith, J. (2021). How to organize creative and innovative teams: creative self-efficacy and innovative team performance. Creativity Research Journal, 33(2), 168-179.

Thanks for allowing me to review your article!

Author Response

Dear Reviewers:

We are thankful for the time and efforts of the reviewers. The review comments are valuable for us to help improve the quality of our manuscript. We have considered and addressed all the comments from the reviewers, and the detailed point-to-point response is provided below. The amendments are highlighted in yellow in the revised manuscript.

Thank you for giving us the opportunity to revise our manuscript.

Yours sincerely,

Reviewer 1

Comment 1: This one discusses the usefulness of creative self-efficacy leading to team members viewing each other as a resource and greater team satisfaction: Swab, R.G., Cogan, A., Pret. T., & Marshall, D.R. (2021). Examining the creative self-efficacy, goal interdependence, and satisfaction of new venture teams in the board game industry. Entrepreneurship Research Journal. doi.org/10.1515/erj-2021-0142

Response 1: Thanks for the valuable comments and suggestions. We have added it to the references. And highlight it in yellow.

Comment 2: This one further supports your argument of creative self-efficacy and innovation: Park, N. K., Jang, W., Thomas, E. L., & Smith, J. (2021). How to organize creative and innovative teams: creative self-efficacy and innovative team performance. Creativity Research Journal, 33(2), 168-179.

Response 2: Thanks for the valuable comments and suggestions. We have added it to the references. And highlight it in yellow.

Reviewer 2 Report

The paper was very interesting to read and it has brought up a new dimension in creative studies. I would like to suggest the followings to enhance the article:

1. In the Introduction, please put more information on the Creative Industries in China; the current situation of the industries and the issues surrounding the professionals.

2. The Discussion of the Findings - practical implications - need more discussion in the context of China, in order for the readers to make sense of the study.

Author Response

Dear Reviewers:

We are thankful for the time and efforts of the reviewers. The review comments are valuable for us to help improve the quality of our manuscript. We have considered and addressed all the comments from the reviewers, and the detailed point-to-point response is provided below. The amendments are highlighted in yellow in the revised manuscript.

Thank you for giving us the opportunity to revise our manuscript.

Yours sincerely,

Reviewer 2

Comment 1: In the Introduction, please put more information on the Creative Industries in China; the current situation of the industries and the issues surrounding the professionals.

Response 1: Thanks for the valuable comments and suggestions. We have added the corresponding Chinese studies to the last paragraph of the "Introduction". It is also highlighted in yellow.

Comment 2: The Discussion of the Findings - practical implications - need more discussion in the context of China, in order for the readers to make sense of the study.

Response 2: Thanks for the valuable comments and suggestions. We have added a discussion of cultural innovation team management in the Chinese context to the practical implications. Please see the yellow highlighted areas of the manuscript for the specific additions.

Reviewer 3 Report

This is a very interesting research topic. I think this article is worthy of publication, but still needs some revisions. My comments are as follows:

1. I suggested to change "Background of the study" to "Introduction".

2. In Background, I suggested to strengthen the overview of China. The last paragraph of Background is suggested to be changed to "Introduction".

3. The last paragraph of Background is suggested to be moved to "Conclusion".

4. In the last paragraph of Background, it is suggested to add the purpose of the study.

5. In the Hypothesis, the number of citations is not enough to confirm the results, please add more.

6. Please add the title and description of "Research Model".

7. Please change the title of 3.1 section to "Procedures and Participants". And describe the sampling technique in this section.

8. Please move the first paragraph of 3.1 section to 3.2 section.

9. For the selection factor of Control variables, please specify in Background or Hypothesis.

10. Please add the introduction of statistical software in chapter 4.

11. Please add the figure of model validation.

12. Please change the chapter title of "Conclusions" to "Discussion" or "Discussion and Recommendation".

13. Please add the section of conclusions.

14. Please add more references for 2023, 2022 and 2021.

Author Response

Dear Reviewers:

We are thankful for the time and efforts of the reviewers. The review comments are valuable for us to help improve the quality of our manuscript. We have considered and addressed all the comments from the reviewers, and the detailed point-to-point response is provided below. The amendments are highlighted in yellow in the revised manuscript.

Thank you for giving us the opportunity to revise our manuscript.

Yours sincerely,

Reviewer 3

Comment 1: 1.I suggested to change "Background of the study" to "Introduction".

Response 1: Thanks for the valuable comments and suggestions. We have corrected it. The title has been changed to “Introduction”.

Comment 2: In the Hypothesis, the number of citations is not enough to confirm the results, please add more.

Response 2: Thanks for the valuable comments and suggestions. We have added to the corresponding hypothesis supporting the literature. The added literature is highlighted in yellow.

Comment 3: Please add the title and description of "Research Model".

Response 3: Thanks for the valuable comments and suggestions. We have modified it and highlighted it in yellow.

Comment 4: Please change the title of 3.1 section to "Procedures and Participants". And describe the sampling technique in this section.

Response 4: Thanks for the valuable comments and suggestions. We have modified it and highlighted it in yellow.

Comment 5: Please move the first paragraph of 3.1 section to 3.2 section.

Response 5: Thanks for the valuable comments and suggestions. We have modified it and highlighted it in yellow.

Comment 6: Please add the introduction of statistical software in chapter 4.

Response 6: Thanks for the valuable comments and suggestions. We have modified it and highlighted it in yellow.

Comment 7: Please change the chapter title of "Conclusions" to "Discussion" or "Discussion and Recommendation".

Response 7:Thanks for the valuable comments and suggestions. We have changed the title of Chapter 5 to "Discussion and Recommendation".

Comment 8: Please add the section of conclusions.

Response 8: Thanks for the valuable comments and suggestions. We have modified it and highlighted it in yellow.

Comment 9: Please add more references for 2023, 2022 and 2021.

Response 9: Thanks for the valuable comments and suggestions. We have added the last three years of literature and highlighted them in yellow.

Comment 10: For the selection factor of Control variables, please specify in Background or Hypothesis.

Response 10: Thanks for the valuable comments and suggestions. We have added the last three years of literature and highlighted them in yellow.

Comment 11:

(1) In Background, I suggested to strengthen the overview of China. The last paragraph of Background is suggested to be changed to "Introduction".

(2) In the last paragraph of Background, it is suggested to add the purpose of the study.

Response 11: Thanks for the valuable comments and suggestions. We have put the "Throughout the existing studies, it can be found that current researchers are increasingly focusing on the motivation and promotion team member's creative behaviors in various industries [55-57]...." Add to the last paragraph of the Introduction. And highlighted them in yellow.

Comment 12: Please add the figure of model validation.

Response 12: Thanks for the valuable comments and suggestions. We have added the model validation results graph as Figure 2 of this study.

Round 2

Reviewer 3 Report

Dear Authors,

You have revised the manuscript and has improved significantly, so I will suggest editor that it can be accepted this revision.

Best regards

Author Response

Dear reviewer, thanks great for your helpful comments and Recognition, which are so important to improve our paper